# Relationship Satisfaction, Attachment, and Perinatal Depression in Women of Color: A Quantitative Investigation

**DOI:** 10.3390/bs14121142

**Published:** 2024-11-28

**Authors:** Reihaneh Mahdavishahri, Dumayi Gutierrez

**Affiliations:** The Professional School of Psychology, Alliant International University, San Diego, CA 92131, USA; dumayi.gutierrez@alliant.edu

**Keywords:** relationship satisfaction, attachment, perinatal depression, women of color, couple therapy

## Abstract

The purpose of this study was to examine the relationships among relationship satisfaction, perceived attachment injury, and perinatal depression for Women of Color (WOC) who have given birth within the last 12 months. In addition, this study aimed to examine the impact of relationship satisfaction and romantic attachment quality on these mothers’ attachments to their newborns. The sample consisted of 120 WOC with perinatal depression. Linear regression and hierarchical multiple regression were used for data analysis. The results indicated that (a) lower relationship satisfaction is predicted with higher severity of perinatal depression, (b) attachment insecurity predicts the severity of perinatal depression, and (c) relationship dissatisfaction, attachment insecurity, and a disrupted attachment bond between mothers and their newborns are predictive of perinatal depression. The results of this study have significant implications for couple and family therapists addressing the needs of pregnant and postpartum WOC. Ultimately, fostering healthy couple relationships during these critical times can play a crucial role in enhancing maternal mental health and overall family wellbeing.

## 1. Introduction

The birth of a child represents a major life cycle transition for mothers, their partners, as well as their families [1]. In 2021, over 3000 births were recorded in the United States and, on average, about 23% of these births were among Latinx mothers, about 15% were among Black mothers, and about 0.8% among Native American mothers [2]. Welcoming a new baby into the world signifies three significant life-altering changes that happen at once, including pregnancy, birth, and becoming a parent [1]. For women, the experience of becoming a mother is often regarded as a joyous occasion [3]; realistically, it can be a highly stressful and challenging transition that leads to permanent shifts in a woman’s identity [4]. The depiction of the transition to motherhood as effortless and natural, along with the celebration of this significant life event across cultures, create expectations that many women may be unable to meet. As a result, many mothers may find themselves struggling with sadness, anxiety, and shame during pregnancy and after childbirth [4].

These psychological challenges are closely associated with the demands of motherhood and the quality of the mothers’ relationship and bonding with their newborn baby(s) after birth [5]. However, the emotional experiences of women during this time cannot be easily generalized; they vary widely and are influenced by numerous factors. These include ethnic and cultural identity, socioeconomic status, support systems, physical health, mental health history, family medical history, and access to resources and healthcare [6,7,8,9,10]. While pregnancy is often portrayed as a joyful and fulfilling life event across various cultures, it is rarely experienced without distress and a storm of emotional and psychological challenges [11].

Among the challenges many mothers face during pregnancy and after childbirth are adverse mental health outcomes such as perinatal depression. Perinatal depression, a mood disorder which can affect women during pregnancy and up to a year after giving birth [5], is the most common complication of pregnancy and birth worldwide and is considered a major public health issue in the United States [12]. One in seven women with a recent birth report experiencing symptoms of perinatal depression [13] such as insomnia or hypersomnia, fatigue, difficulty with concentration, ongoing anxiety, and thoughts of suicide or infanticide in severe cases [5]. Women of Color (WOC) are three times more likely than White women to experience perinatal depression, with Native American mothers having the highest rate of reported perinatal depression among all racial and ethnic groups in the United States [12]. These experiences are often dismissed, minimized, or misdiagnosed as postpartum blues, preventing many pregnant and postpartum women from receiving the support they need [3]. Notably, the literature has maintained an individualistic lens with pregnancy and childbirth, neglecting the vital relational aspects of this life event for many women and the romantic partners involved [3].

Pregnancy and childbirth bring exceptional circumstances in a couple’s relationship, accompanied by anticipated and unforeseen challenges and demands [11]. The inherent fragility that pregnancy and childbirth entail make couples more susceptible to encountering relational distress, and potentially even injury to their attachment bond [14]. Attachment refers to a complex behavioral and psychological system that drives us to seek support from others in times of distress [15]. Research has revealed that relationship satisfaction significantly influences the quality of attachment between partners [14,16]. These findings emphasize the crucial role of secure relationships in fostering healthy emotional connections. However, little is known about the experiences of WOC and ways in which their attachment with their romantic partners may negatively or positively impact their wellbeing during pregnancy, postpartum, and through perinatal depression [17].

Thus, this study highlights significant relationships between relationship satisfaction and attachment between partners and their impact on negative health outcomes, such as perinatal depression in mothers, through an attachment-informed lens.

## 2. Perinatal Depression

Previous research has often identified and referred to depressive symptoms related to pregnancy and birth as postpartum depression, reflecting the recognition of these symptoms post-birth [3]. In light of recent research, which addresses the experiences of women not just after birth but during pregnancy [18], this paper will use perinatal depression when discussing the symptomology of depressive symptoms during pregnancy and after birth. Perinatal depression remains among the most undetected and untreated mental health conditions affecting women [1], with devastating consequences for mothers, their families, and newborn babies [18]. Among the consequences are the impacts of perinatal depression on maternal attachment and the adverse health outcomes for children [3,19]. Newborn babies of mothers with perinatal depression are more prone to colic and diarrhea and are at increased risk of developing asthma, cardiovascular disease [3,20], and dysregulated affect and behaviors [16]. These adverse health outcomes persist into later developmental stages, as affected children encounter difficulties with social skills and forming secure attachments with peers, show sign of cognitive delays, and report symptoms of mental health conditions such as anxiety and depression as young adults [3]. Despite longitudinal outcomes, less than 10% of reported women struggling with perinatal depression receive proper care; this is even lower for WOC, with less than 4% receiving the support they need [4,18].

Social and psychological factors can significantly affect the onset and severity of perinatal depression, its prognosis, and its treatment outcomes [1]. While some studies have argued that the prevalence rate of depression postpartum is similar to that of women in general [21,22], there is overwhelming evidence suggesting that pregnant and postpartum WOC are at a higher risk of developing peripartum depression [4]. Studies that claim the prevalence rate of depression in perinatal periods and other periods of women’s lives are similar remain limited to data which have excluded the experiences of WOC. Recent research, albeit limited, indicates that pregnancy and motherhood are particularly vulnerable times for WOC and increase their risk of developing perinatal depression. About 40% of WOC experiencing a major depressive episode will experience their initial episode during pregnancy, and up to 50% of WOC will experience moderate to severe depressive symptoms after giving birth for the first time [13,23]. This provides further evidence that pregnant and postpartum WOC are at increased risk of developing perinatal depression.

### 2.1. Adverse Health Outcomes for Children

Prenatal maternal stress has been identified as a significant risk factor for the onset and severity of peripartum depression [1]. The relationship between prenatal maternal stress and adverse health outcomes for children has been well established [3,19]. Negative health outcomes for children of mothers struggling with perinatal depression are associated with mothers’ difficulty to properly care for their newborn babies [20]. Identified factors such as anhedonia, exhaustion, difficulty with concentration, and depressed mood have been reported to interfere with healthy caretaking behaviors [20]. In addition, mothers who suffer from perinatal depression are more likely to miss important well-child visits and immunizations and may not seek medical care for their children as frequently [3]. This can have negative effects on the overall health and wellbeing of children, as research has shown that maternal depression during the perinatal period can disrupt children’s cardiovascular functioning, increase their risk of gastrointestinal infections, and lower their respiratory health [19]. Therefore, addressing perinatal depression is not only crucial for the health and wellbeing of mothers but also for their children.

### 2.2. Infant-Mother Attachment

Attachment theory, originally developed by Bowlby [15] and expanded by Mary Ainsworth [24], is an interpersonal theory of human development that seeks to examine and explain the dynamics of interpersonal relationships and attachment to a primary caregiver. Infants are biologically programmed to form attachments with their caregivers, particularly their mothers [3,15]. The first 4 months of an infant’s life is a significantly sensitive time to form a secure attachment with their primary caregiver, often their mother [15,24]. The essence of attachment between an infant and their mother involves the reciprocal behaviors in which they both engage, such as eye contact or gazing, positive affect, vocalization, and play [1,15]. Thus, the mother–infant relationship is arguably the most significant relationship for the infant and represents a major psychological process for both the mother and the child [3].

The quality of the attachment bond formed between child and mother directly impacts the child’s mental and physical wellbeing [1,24]. When a mother is inattentive, unavailable, and unresponsive to the child’s needs, fear-based responses are triggered in the child. Such consistent unresponsiveness can disrupt the child’s cognitive, emotional, social, and behavioral systems and shape their internal working model of the self around insecure-attachment-oriented behaviors [1,3,24]. It is important to note that while attachment-oriented behaviors are typically activated post-birth, bonding processes that set the stage for attachment begin during pregnancy [3,15]. Research has shown that depression during pregnancy can adversely impact the unborn child, and that affect dysregulation in newborns of mothers with perinatal depression is thought to have origins in the prenatal period [1,3]. Thus, the quality of the attachment bond formed between mother and child during the perinatal period is crucial for the child’s mental and physical health, with disruptions in responsiveness leading to maladaptive engagement and insecure-attachment-oriented behaviors [1,3]. Depression during pregnancy may adversely affect the unborn child’s development, emphasizing the importance of the bonding process that begins during pregnancy.

### 2.3. Attachment Between Partners

Pregnancy and birth present new challenges for a relationship and bring about unique life cycle transitions for the couple and the family awaiting the arrival of a new member [16]. Existing relationship challenges and stressors are compounded by the changes associated with pregnancy and birth, such as relationship dynamics, shifts in parenting roles, the negotiation of new roles and responsibilities, and the financial burdens of a new child [16]. There have been clear links found between perinatal depression and relationship satisfaction [16]. In fact, relationship distress and dissatisfaction have been identified as risk factors for perinatal depression [17]. Relationship quality is also associated with the severity, duration, and relapse rate of perinatal depression symptoms [11]. Further, women in unsupportive relationships are more likely to have more severe depressive symptoms, have more difficulty recovering from these symptoms, and are at increased risk of having chronic depression following pregnancy and birth [11,16]. Thus, women’s attachment to their romantic partners, as indicated by the quality of the partnership, relationship satisfaction, and support, impacts and is impacted by perinatal depression [16]. However, there are significant gaps in the literature when it comes to experiences of WOC during pregnancy and postpartum and how their romantic relationships impact the course of their peripartum depression and recovery.

#### Attachment Injury

Pregnancy and birth embody unique moments in a couple’s relationship, with expected and unexpected challenges and demands [11]. Given the vulnerability that pregnancy and birth invite into relationships, couples are at increased risk of experiencing relationship distress and potentially an attachment injury [3,14,25]. Attachment injury is defined as a pivotal moment in the romantic relationship, during which one partner is perceived to be unavailable, emotionally disengaged, and unreachable during a particularly vulnerable time, shifting the perception of the partnership to one that is fundamentally unreliable [14,25]. On the other hand, partners consistently demonstrating a secure attachment bond, which is evident through their mutual emotional accessibility, engagement, and responsiveness, navigate the challenges of pregnancy and birth more easily [14]. Women in these relationships are less likely to report symptoms of perinatal depression and are more likely to recover from reported symptoms completely [26]. Thus, the quality of the attachment bond between partners can either protect against or increase the risk of perinatal depression during and after pregnancy [27]. However, the impact of the attachment bond between WOC and their romantic partners on their pregnancy and postpartum experiences remains unknown.

This study aims to bridge the current gaps in the literature by examining the relationship between perinatal depression and attachment among WOC. Through an attachment lens and a quantitative design, this study surveyed the experiences of WOC with perinatal depression in romantic relationships during pregnancy and within the first year after birth. The purpose of this investigation was to examine the relationship between perinatal depression and quality of attachment to their romantic partners for WOC, as well as level of attachment insecurity and its relationship to perinatal depressive symptoms. Further, this study examined the impact of partner attachment on the quality of newborn attachment for WOC. The results of this study will help researchers gain a better understanding of the relational aspects of perinatal depression among WOC and advocate for more culturally informed and collective treatment options for this population.

## 3. Method

This quantitative study used primary data to examine the relationship between attachment insecurity (anxious attachment and anxious avoidance), mother–infant bonding, and perinatal depression among WOC and if attachment between partners (relationship dissatisfaction) predicts perinatal depression. Linear and hierarchical multiple regression models were utilized to examine the statistical power of attachment in predicting perinatal depression among WOC. This study was approved by the Institutional Review Board. The following hypothesis were tested:

**H1.** *Relationship dissatisfaction between partners will predict perinatal depression*.

**H2.** 
*Relationship dissatisfaction and attachment insecurity (anxious attachment and anxious avoidance) between partners will predict mother–infant bonding disturbances.*


**H3.** 
*Attachment insecurity between partners will predict perinatal depression.*


### 3.1. Participants

The participants of this study were WOC in romantic relationships struggling with symptoms of perinatal depression. The term WOC refers to women who identify as non-White, including those identifying as Latinx, American Indian or Alaska Native, Black or African American, Asian, Native Hawaiian or Pacific Islander, Middle Eastern and/or North African, East Asian, South Asian/Indian, Filipino, and multiethnic. All participants were recruited using a survey panel service (Alchemer) and completed an anonymous online survey via this server. Alchemer is an online research company that provides access to nationwide samples [28]. Women were able to participate in the study if (1) they were 18 years or older; (2) self-identified as WOC; (3) were in a relationship with their partner; (4) struggled with symptoms of perinatal depression; (5) had experienced perinatal depression symptoms during pregnancy and/or up to 1 year after birth; and (6) were fluent in reading and writing in English. WOC reporting symptoms associated with postpartum psychosis such as hallucination, paranoia, and delusion were excluded from this study. Based on estimates from a priori G*power analysis [29], a sample size calculation strategy utilized to calculate the sample size necessary for a hierarchical multiple regression analysis, the targeted sample size determined accounted for (alpha = 0.05, effect size = 0.15, predictors = 3) a power of 0.95. The sample size reached for this study was (n = 120) individuals.

On average, participants were in their late 20s (M = 29.8, SD = 6.8). The majority reported the onset of perinatal depression after birth (n = 85, 70.8%), with 71% not having received any treatment for their symptoms. Over half of participants identified as Black or African American (n = 67, 55.8%), with 16.6% (n = 20) identifying as multiethnic. Over 80% of participants identified as heterosexual (n = 98). The majority of the participants’ partners were present during the delivery of their child (n = 95, 79.2%). In total, 67.5% (n = 81) of the participants indicated receiving higher education (some college and higher), with over half of participants indicating an annual household income of USD 35,000 to over USD 100,000 (n = 63, 52.5%) (see Table 1).

### 3.2. Measures

#### 3.2.1. Demographic Information

The initial questionnaire gathered demographic information such as age, race, ethnicity, level of education, sexual identity, duration of reported struggle with symptoms of perinatal depression, onset of perinatal depression symptoms (during pregnancy and/or after birth), whether they had received any formal diagnosis and treatment for perinatal depression, and the type of treatment they had received (individual therapy, couple therapy, and/or pharmacological interventions).

#### 3.2.2. Edinburgh Postnatal Depression Scale (EPDS)

To evaluate the severity of symptoms associated with peripartum depression, the EPDS was used [30]. The EPDS (Cronbach’s alpha of 0.8) addresses perinatal depression symptoms by measuring the severity of symptoms in pregnant mothers and those who experience depressive symptoms after giving birth. This brief questionnaire has been shown to be a valid and reliable measure in detecting and measuring the severity of perinatal depression symptoms across diverse populations and is widely used in current research and treatment of perinatal depression [30]. This 10-item questionnaire evaluated the participants’ responses on a four-point Likert scale rated from 0 (never) to 4 (most of the time) over the past 7 days before taking the measure (e.g., “I have looked forward with enjoyment to things”).

#### 3.2.3. The Experiences in Close Relationships–Relationship Structures (ECR-RS)

ECR-RS is a 9-item self-report measure of adult attachment in a romantic relationship [31]. This measure is designed to evaluate patterns of attachment in close relationships, in this case with a romantic partner, and contributes to our understanding of potential attachment injuries in romantic relationships. Six of the nine items measure attachment avoidance (e.g., “I don’t feel comfortable opening up to my partner”) (a = 0.94), and three of the nine items measure attachment anxiety (e.g., “I am afraid that my partner may abandon me”) (a = 0.90), with higher scores in each category indicating an attachment insecurity between partners. Responses were recorded on a Likert scale of 1 (strongly disagree) to 7 (strongly agree). Based on an online sample of over 21,000 participants, ECR-RS scores are shown to be reliable and emerging cross-cultural studies point out its validity [31,32].

#### 3.2.4. The Revised Dyadic Adjustment Scale (RDAS)

To examine the attachment bond between the mothers and their partners, the RDAS was utilized. The RDAS is a 14-item measure which tests relationship satisfaction and individuals’ perceptions of their romantic partnerships (e.g., “do you ever regret that you married or lived together?”) [33]. The RDAS has been shown to be a reliable and valid measure of relationship satisfaction with a reported internal consistency of a = 0.90 [34]. The 14 items were scored on a Likert scale and participants’ responses were summed to create a total score ranging from 0 to 69. Higher scores indicated relationship satisfaction and more positive dyadic adjustment, while lower scores on the scale indicated relationship dissatisfaction and more negative dyadic adjustment.

#### 3.2.5. Postpartum Bonding Questionnaire (PBQ)

The BPQ is a self-report measure that assesses mothers’ attitudes and emotions regarding their newborn infants [35]. The 25 items were scored on a six-point scale with responses ranging from 0 (never) to 5 (always) (e.g., “I resent my baby”). Higher scores indicate negative attitudes and affect towards the baby and are indicative of disturbances in mother–infant bonding. There is strong evidence for reliability and cross-cultural validity of this measure, and it is used in clinical settings and research [36].

### 3.3. Analytic Strategy

Data analysis in this study utilized Statistical Package for the Social Sciences (SPSS 28) software and was conducted in various steps. First, descriptive data were cleaned and coded from descriptive demographic answers in Alchemir (see Table 1) and means and standard deviations were used to provide descriptive information. Alpha reliabilities were calculated to determine the internal consistency of each measure (see Table 2). Second, correlations were run between independent and dependent variables (Table 3). Third, a linear regression was run to test if relationship dissatisfaction predicted perinatal depression (Table 4). Fourth, a multiple hierarchical regression model was run to examine the relationship between relationship dissatisfaction, attachment insecurity, and mother–infant bonding disturbances (Table 5). Additionally, this study used a multiple hierarchical regression model to examine the relationship between relationship dissatisfaction, attachment insecurity, and mother–infant bonding disturbances, with prenatal depression as the outcome variable (Table 6). This study employed multiple regression analysis to determine the extent to which the independent variables (relationship dissatisfaction, attachment insecurity, and mother–infant bonding disturbances) explain the variance in the dependent variable of perinatal depression at a significant level. Additionally, this study used hierarchical multiple regression analysis to examine the relative predictive significance of the independent variables, with the order of entry based on theoretical considerations and literature evidence determined by the researchers [6].

According to attachment theory and the theory of adult love, a secure attachment is a protective factor against adverse mental health outcomes while insecure attachment increases the risk of adverse mental health outcomes such as perinatal depression. Therefore, relationship dissatisfaction was entered into a hierarchical regression predicting perinatal depression for mothers. Attachment insecurity (anxious attachment and anxious avoidance) was entered in the model after relationship dissatisfaction. It is reported in the literature and the findings of the stress-buffering model that marital dissatisfaction can reduces a person’s perception of the support that is available to them, thus increasing the risk of experiencing an attachment injury in the relationship and therefore attachment insecurity [6]. As a result, attachment insecurity was entered into the model after relationship dissatisfaction. Anxious attachment and anxious avoidance each predict attachment injury as they each contribute to the attachment injury model separately. Mother–infant bonding disturbances were entered next in the model, as perceived relationship quality is a contributing factor to mother–infant interactions.

## 4. Results

### 4.1. Relationship Dissatisfaction Will Predict Perinatal Depression

A linear regression analysis for relationship dissatisfaction predicting perinatal depression was used to test the first hypothesis. Relationship dissatisfaction significantly predicted higher levels of perinatal depression (*R*^2^ = 0.080, *F*(1, 118) = 10.312, *B* = −0.133, *p* = 0.002), with relationship dissatisfaction accounting for 8% of the variance (Table 4).

### 4.2. Relationship Dissatisfaction and Attachment Insecurity (Anxious Attachment and Anxious Avoidance) Between Partners Will Predict Mother–Infant Bonding Disturbances

Multiple hierarchical regression analysis was used to test the second hypothesis. Model 1 showed that relationship dissatisfaction was not a significant predictor of mother–infant bonding disturbances (*B* = −0.109, *F*(1, 118) = 1.16, *p* = 0.282). The model accounted for 1% of the variance in mother–infant bonding (*R*^2^ = 0.010). Model 2 showed that the overall model was not statistically significant in predicting mother–infant bonding disturbances (*B* = −0.045, *F*(3, 116) = 0.0736, *p* = 0.533). The model accounted for 2% of the variance in mother–infant bonding (*R*^2^ = 0.019) after controlling for the effects of relationship dissatisfaction and attachment insecurity (attachment anxiety and attachment avoidance). Therefore, the results indicate that relationship dissatisfaction and attachment insecurity between partners are not significant predictors of mother–infant bonding disturbances for WOC (Table 5).

### 4.3. Attachment Insecurity Between Partners Will Predict Perinatal Depression

A four-stage hierarchical multiple regression was conducted with perinatal depression as the dependent variable to test hypothesis three. Relationship Dissatisfaction was entered at stage one of the regression, as it is a well-documented predictor of mental health challenges during the perinatal period [3,11,16]. Attachment insecurity variables (Anxious Attachment and Avoidant Attachment) were entered at stage two, given the central role these attachment styles play in emotional regulation and relational dynamics [17,27,31] and Mother–Infant Bonding Disturbances at stage three. The variables were entered in this order as it seemed chronologically plausible given couples’ attachment patterns and quality precede mother–infant bonding during pregnancy and following birth (Table 6).

As predicted, the hierarchical multiple regression revealed that at stage one, Relationship Dissatisfaction contributed significantly to the regression model (*R^2^* = 0.080, *F*(1, 118) = 10.31, *β* = −0.283, *p* = 0.002) and accounted for 8% of the variation in Perinatal Depression. Introducing the Attachment Insecurity variables (Anxious Attachment and Anxious Avoidant) explained an additional 10.5% of variation in Perinatal Depression and this change was significant (*R^2^* = 0.185, *F*(2, 116) = 7.468, *β* = 0.430, *p* < 0.001). Adding Mother–Infant Bonding to the regression model explained an additional 15.8% of the variation in Perinatal Depression and this change was significant (*R^2^* = 0.344, *F*(4, 115) = 15.045, *β* = 0.586, *p* < 0.001).

## 5. Discussion

As discussed, research lacks an exploration of the experiences of WOC and how attachment with their romantic partners might impact their mental wellbeing during pregnancy and postpartum, as well as the attachment bonds they form with their newborns. This study fills these gaps, and the results found support for two of the three hypotheses examined. Specifically, the findings emphasize the essential role of partnerships in influencing the mental health outcomes of WOC during this pivotal time. This underscores the necessity for further exploration of these dynamics and the tailoring of our relational interventions to address their needs.

As hypothesized, relationship satisfaction was found to have a significant inverse relationship with perinatal depression for WOC. The more satisfied the participants were in their relationship with their romantic partners, the lower the reports of perinatal depression symptoms. Higher relationship satisfaction is reported to indicate more secure attachment between partners and has been identified as a protective factor against adverse mental health outcomes including depression during pregnancy and postpartum [16], though the experiences of WOC were overwhelmingly excluded from such reports. The results of this study are supported by literature that indicates strong links between perinatal depression and relationship dissatisfaction [11,16] and bridges the gap regarding the experiences of WOC in romantic relationship during pregnancy and postpartum. The unique circumstances of pregnancy and childbirth create many challenges and demands for couples, both expected and unexpected [11]. These experiences can leave couples vulnerable to experiencing relationship difficulties. Relationship satisfaction, measured in various aspects of a romantic partnership such as communication, intimacy, and overall sense of happiness, provides a window into the workings of couples’ attachment and relational wellbeing [33]. The results of this study provide further evidence for the significance of the quality of a romantic partnership in protecting against adverse mental health outcomes during pregnancy and postpartum, particularly for WOC, and the additional support that secure attachment can offer to protect against perinatal depression.

In addition to relationship dissatisfaction, attachment bond ruptures and injuries with romantic partners also have significant implications for the mental wellbeing of mothers and their newborns. Depression occurring during the peripartum period can lead mothers to feel isolated and overwhelmed, as they may struggle to form a close bond with their newborns, leading to increased feelings of inadequacy and worthlessness [18,27]. As a result, these mothers often rely on their partners for additional support [16]. However, underlying insecure attachment patterns and relationship difficulties can expose these mothers to more isolation and ruptured attachment with their partners [16,27]. Research is lacking on the links between attachment bond injuries and adverse mental health outcomes for women during pregnancy and postpartum, with no such links examined among WOC. The quality of the attachment bond between partners has the power to function as a source of emotional security or vulnerability, significantly influencing women’s wellbeing during critical phases of pregnancy and childbirth [14]. Adverse health outcomes of disrupted attachment bonds formed between mothers and their newborns have long been established in the literature, with results demonstrating the pivotal role of this relationship and its long-lasting impact on children’s development and wellbeing well into adulthood [20]; however, such studies have failed to include the experiences of WOC, and the nuances of their attachment bonds formed with their newborns.

The results of this study found attachment bond insecurity to be more closely associated with mother–infant bonding disturbances than relationship dissatisfaction, even though this association was not found to be significant. Given that an attachment injury is typically characterized as a moment of intense isolation and abandonment by the injured partner, which can fundamentally alter their internal working models of self and others, thus increasing attachment insecurity [25], the results highlight the relationship between such a fundamental shift and its impact on other aspects of these mothers’ lives, including their bond with their children. While this study did not find strong support for attachment insecurity and relationship dissatisfaction to be significant predictors of mother–infant bonding disturbances among WOC, future research would benefit from exploring specific factors impacting these bonds. Considering the significance of birth for families from collectivistic cultures, the results of this study suggest that other factors, such as familial support outside of the romantic partnership, may act as protective measures. Thus, the disruption of attachment between mothers and their infants could be prevented. While this study did not find attachment insecurity and relationship dissatisfaction to be significant predictors of mother–infant bonding disturbances, future research would benefit from exploring specific factors impacting mother–infant bonding among WOC.

This study did observe a significant relationship between attachment bond insecurity and peripartum depression among WOC. Both anxious attachment and anxious avoidance patterns, which are indicative of attachment bond injuries between romantic partners, strongly correlated with perinatal depression and this relationship was found to be significant. As a result, even though attachment insecurity and relationship dissatisfaction were not found to predict mother–infant bond disturbances, they did predict peripartum depression as suggested by previous research. Thus, these findings provide valuable information that adds to the limited existing knowledge regarding the experiences of WOC and their romantic relationships, and how these experiences can impact their mental health.

## 6. Clinical Implications

The field of therapy has historically been shaped by ideologies centered on individualistic values. Consequently, perinatal depression is often assessed and treated in isolation, overlooking the relational dimensions of women’s experiences during pregnancy and postpartum. As couple and family therapists, we have a unique opportunity to challenge and reformulate the pathologizing clinical approaches to conditions affecting women. By exploring family dynamics, relationship quality, and how these relational factors influence maternal wellness, we can shift the therapeutic focus from individual interventions to more systemic and relational approaches.

There is also a significant gap in the literature regarding effective treatments for WOC experiencing perinatal depression [3,7]. The majority of the studies on perinatal depression have been completed with White, married, and educated women from higher socioeconomic backgrounds, further widening the current gap in our knowledge about the unique experiences of WOC [8,9]. The majority of women who initiate services for their symptoms associated with perinatal depression are provided with pharmacological interventions, with few non-medication treatment options available to them [1,10,23]. In fact, WOC are less likely to initiate care for perinatal depression due to concerns about medication and pharmacological side effects

Lack of systemic, family-oriented, religious, and spiritual-based services are among common reasons why WOC do not seek services or fill out prescribed medication [23,37,38,39], despite growing evidence on the relational aspects of this condition, including the different ways in which it impacts and is impacted by the relationship between partners, the family dynamic, and the relationship between the mother and her newborn infant(s) [10,11,40]. Thus, the current individual-focused approaches to treating perinatal depression are insufficient. It is time to advocate for systemic interventions that acknowledge the importance of relationships in women’s mental wellbeing, particularly during the critical phases of motherhood. The findings from this study, particularly regarding the significant role of relationship satisfaction and attachment security, highlight the need for such systemic interventions that address these relational dynamics. Clinical practices, such as couples counseling and attachment-based interventions, could be pivotal in supporting WOC by strengthening relational bonds and reducing the risk of perinatal depression. By incorporating relationship-focused care into perinatal mental health services, we can better meet the needs of WOC and improve outcomes for mothers and their families.

## 7. Limitations

In March of 2020, the WHO (2022) declared the coronavirus infection (COVID-19) as a global pandemic. At the time this study was proposed, COVID-19 (SARS-Cov-2) had claimed the lives of over 6 million people worldwide [41]. There have been a few studies conducted worldwide on the impacts of the COVID-19 pandemic on the experiences of women during pregnancy and postpartum [42,43]. However, there are conflicting reports on the ways in which COVID-19 may have impacted the experiences of these women, with some studies reporting higher rates of anxiety, depression, and peripartum symptoms [42], and others reporting no significant and/or lasting effects [43]. While this study acknowledges the significance of the COVID-19 pandemic on the experiences of its participants during pregnancy and postpartum, investigating the relationship between COVID-19 and peripartum depression for WOC was beyond the scope of this study.

This study utilized a quantitative methodology to examine the relationship between attachment and perinatal depression among WOC. Given that cultural and ethnic identity are complex and multifaceted phenomena, the nuances of participants’ experiences may be difficult to measure quantitatively. Capturing the unique experiences of WOC with perinatal depression and any cultural differences in the presentation of perinatal depression may be better achieved through phenomenological studies. This study also clustered several cultural identities in its findings. Clustering multiple cultural identities may oversimplify the complexities of each individual culture and fail to fully capture the unique experiences and perspectives of each cultural group. It may also homogenize and erase important differences and variations within and across cultural groups, excluding certain cultural groups that do not fit neatly into the identified clusters.

Another limitation of this study was utilizing hierarchical regression modeling for data analysis. Like any regression technique, hierarchical regression makes certain assumptions about the order in which independent variables are entered in the model [44]. While review of the literature and theory informed the researcher’s decision to enter the variables into the model, there may be potential bias inherent in theories selected for this research. In addition, this study’s cross-sectional and correlational design introduces several limitations [45]. Given the nature of these designs, causal relationships between the variables—such as relationship satisfaction, attachment insecurity, or perinatal depression—cannot be established. Correlational studies are limited to identifying the strength and direction of relationships between variables at a single point in time, without revealing the underlying causes or accounting for other potential influences [45]. Furthermore, this study does not capture other factors that could significantly affect these outcomes, such as access to mental health services, social support systems, or cultural influences, all of which may play a critical role in shaping the experiences of WOC during the perinatal period.

## 8. Future Research

In light of the limitations discussed, future research could address these gaps by utilizing longitudinal designs to examine perinatal depression at multiple time points, allowing for a deeper understanding of how relationship satisfaction and attachment insecurity evolve over the course of pregnancy and the postpartum period. Additionally, incorporating dyadic data from both partners could provide valuable insights into dual attachment styles and the shared experiences of pregnancy and postpartum challenges, further enhancing our understanding of how these dynamics influence maternal mental health.

Future research could benefit from integrating the biopsychosocial model to gain a more nuanced understanding of perinatal depression. This approach would examine not only psychological and relational factors, but also biological influences (such as hormonal changes, genetics, and physical health) and social determinants (including cultural norms, socioeconomic status, and social support). The goal is to address gaps in the existing literature and identify unique risk and protective factors within diverse communities, with a particular focus on WOC. By broadening the scope in this way, research can offer a more comprehensive understanding of perinatal depression, which is crucial for developing targeted interventions and support systems that better meet the needs of these populations.

Sexual and gender minority (SGM) individuals may be at increased risk of perinatal depression due to added stressors such as systemic oppression and discrimination, lack of support from family, and legal barriers related to adoption and custody [46]. Additionally, limited research on the experiences of trans men who experience pregnancy after their transition suggests that they may also be at increased risk of developing perinatal depression due to lack of support from family, negative interactions with care providers, and lack of access to gender-affirming services [46]. However, there are significant gaps in the literature when it comes to experiences of lesbian, gay, bisexual, trans, and queer WOC with perinatal depression. There is a significant need to investigate the intersectionality of race, ethnicity, gender identity, and sexual identity and the onset and severity of perinatal depression that future studies need to address.

## Figures and Tables

**Table 1 behavsci-14-01142-t001:** Descriptive demographics of participant sample (n = 120).

Characteristics	Mean	SD	%
Age	29.82	6.88	
Education Level			
Less than a high school diploma			6.7
High school degree or equivalent (e.g., GED)			25.8
Some college, no degree			24.2
Associate degree (e.g., AA)			14.2
Bachelor’s degree (e.g., BA)			20.8
Master’s degree (e.g., MA)			5
Professional degree (e.g., MD)			0.8
Doctorate degree (e.g., PhD)			2.5
Household Income			
Less than USD 20,000			30
UDS 20,000 to USD 34,999			17.5
USD 35,000 to USD 49,999			18.3
USD 50,000 to USD 99,999			10
USD 75,000 to USD 99,999			10
Over USD 100,000			14.2
Onset of Prenatal Depression			
During pregnancy			29.2
After birth			70.8
Received PPD Treatment			29.2
Partner present during delivery			79.2
Sexual identity			
Heterosexual			81.7
Asexual			4.2
Bisexual			6.7
Gay/Lesbian			1.7
Queer			0.8
Other			5
Ethnicity			
Latinx			15.8 4.3 12.5 55.8 1.7 3.3 16.6 0.8 1.7 3.3
American Indian or Alaska Native			
Asian			
Black or African American			
Native Hawaiian or Other Pacific Islander			
Middle Eastern and/or North African			
Multiethnic			
East Asian			
South Asian/Indian			
Filipino			

**Table 2 behavsci-14-01142-t002:** Means, standard deviations, and reliability coefficients for measures.

Variable Measure	*M (SD)*	Alpha
Edinburgh Postnatal Depression Scale (EPDS) [30]	13.14 (5.6)	0.846
The Experiences in Close Relationships–Relationship Structures (ECR-RS)—Attachment Anxiety [31]	16.2 (8.94)	0.902
The Experiences in Close Relationships–Relationship Structures (ECR-RS)—Attachment Avoidance [31]	8.8 (4.9)	0.882
The Revised Dyadic Attachment Scale (RDAS) [33]	39.56 (11.64)	0.887
Postpartum Bonding Questionnaire (PBQ) [35]	39.56 (11.64)	0.902

**Table 3 behavsci-14-01142-t003:** Correlations between predictor variable and prenatal depression (N = 120).

Variables	1	2	3	4	5
1. Prenatal Depression					
2. Attachment Insecurity (Anxious Attachment)	0.332 **				
3. Attachment Insecurity (Anxious Avoidance	0.423 **	0.648 **			
4. Relationship Satisfaction	−2.83 **	−0.793 **	−0.592 **		
5. Mother–Infant Bonding	0.451 **	−0.239 **	−0.177	0.228 *	−0.147

Note: ** *p* < 0.01; * *p* < 0.05.

**Table 4 behavsci-14-01142-t004:** Linear regression analysis for relationship dissatisfaction predicting prenatal depression (n = 120).

			95% CI			
Variable	*B*	*SE B*	*LL*	*UL*	*β*	*p*
Relationships Satisfaction	−0.133	0.041	−0.214	−0.051	−0.283	0.002 **
*R* ^2^		0.080				

Note: ** *p* < 0.05.

**Table 5 behavsci-14-01142-t005:** Summary of hierarchical regression analysis for variance predicting mother–infant bonding disturbances (n = 120).

		Model			Model	
		1			2	
Variable	*B*	*SE B*	*β*	*B*	*SE B*	*β*
Relationship Satisfaction *	−0.109	0.101	−0.099	−0.045	0.169	−0.041
Anxious Attachment				−0.146	1.349	−0.017
Anxious Avoidance				0.823	0.833	0.121
*R* ^2^		0.010			0.019	
*F* for change in *R*^2^		1.168			0.736	

Note: relationship satisfaction was measured using the RDAS, where lower scores indicate greater dissatisfaction. * *p* < 0.05.

**Table 6 behavsci-14-01142-t006:** Summary of hierarchical regression analysis for variables predicting prenatal depression.

Variable	*β*	*t*	*R*	*R* ^2^	Δ*R*^2^
Step 1 Relationship Satisfaction	−0.283	−3.211 *	0.283	0.080	0.080
Step 2 Relationship Satisfaction Anxious Attachment Anxious Avoidance	0.023 0.116 0.362	0.163 0.785 3.242 *	0.430	0.185	0.105
Step 3 Relationship Satisfaction Anxious Attachment Anxious Avoidance Mother-Infant Bonding	0.039 0.123 0.313 0.402	0.312 0.923 3.1 * 5.266 **	0.586	0.344	0.158

Note: * *p* < 0.05; ** *p* < 0.001.

## Data Availability

The link to datasets analyzed during this study is available upon request from the authors.

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
