# Peer review of "Relationship Satisfaction, Attachment, and Perinatal Depression in Women of Color: A Quantitative Investigation"

_behavsci, 2024, doi:10.3390/bs14121142_

Round 1
Reviewer 1 Report
Comments and Suggestions for Authors
Thank you so much for the opportunity to review your work. I think it is an essential contribution to the literature for multiple reasons. Primarily, it centers on the experiences of women of color during a time that is important for them, namely pregnancy and childbirth.
Introduction
* It might be nice to talk about differential experiences of pregnancy and childbirth between women of color and White women more broadly than you currently do. On line 37, you immediately jump into perinatal depression; your argument could be strengthened if you expand on it. I don't think you have to do it, but it may be nice to have additional context.
Methods
* Were any women removed from the data?
* Were the women supposed to have depression during and after pregnancy to be eligible?
* For the EPDS, it would be nice to have a sample item
* For the EPDS, saying the instrument is valid and reliable without providing a specific citation to that point is not very helpful
* Are you measuring peripartum depression or only postpartum? It isn't very confusing with this instrument
* What do the scores on this instrument mean (e.g., higher scores mean more severe symptomatology)?
* For the RDAS, please add a sample item(s)
* For the PBQ, please add sample item(s)
* Consider moving the section about attachment theory and adult love (line 292) to the introduction. It has a strange flow with its current location
Results
* The statistical language needs to be italicized in parentheses (e.g., Line 312[R2 = .080, F(1, 118) = 10.312, β = -.283, p = .002]).
* Why not report the unstandardized coefficient (i.e., B) from Table 4 in line 312?
* The written report (line 312) appears to contradict what is in the table. Unless you are measuring dissatisfaction and calling it satisfaction. Clarifying this will be very helpful
* Line 319 you might consider replacing "address" with "test"
* Reporting the unstandardized Beta might be helpful in your results section. You immediately report R2 , which is fine, but when we consider regression analyses, we use the same y = mx + b formula, and you are not reporting the coefficient in that formula. Just a thought
Discussion
* Consider the cross-sectional and correlational nature of your study in your limitations
* Consider the limitations you all listed and discuss how future research and researchers could fill the gaps you are suggesting now exist in your Future Research section
* Do you think treatment has any effect?
Reviewer 2 Report
Comments and Suggestions for Authors
Review: Relationship Satisfaction, Attachment and Peripartum 2 Depression in Women of Color: A Quantitative Investigation
The title captures the main variables (relationship satisfaction, attachment, and peripartum depression) and specifies the focus on "women of color." However, the phrase "Quantitative Investigation" could be more informative if it briefly specifies the methodology, such as "A Cross-Sectional Quantitative Analysis" or "A Survey-Based Quantitative Study," which can help readers immediately understand the research approach.
Terminology: For accessibility and precision, consider specifying "peripartum depression" to distinguish it clearly from postpartum depression, if that is relevant to your study.
Since "women of color" encompasses diverse backgrounds, the abstract could benefit from specifying any particular ethnic or cultural groups if they are central to the study. This can enhance the relevance and depth of your findings.
Theoretical Framework: Briefly indicating any psychological or sociocultural theories guiding the research could clarify the abstract's focus and intellectual contribution.
Innovation Suggestions:
To make the research more innovative and impactful, consider these angles:
- Intersectionality: Emphasize an intersectional approach by analyzing how relationship satisfaction and attachment styles might intersect with cultural and social pressures unique to different ethnic groups. This could provide nuanced insights into how cultural expectations and family structures affect mental health.
- Biopsychosocial Approach: Integrate a biopsychosocial model to examine not only psychological but also biological and social factors influencing peripartum depression in women of color. This approach can reveal unique risk factors or protective mechanisms present within specific communities.
- Comparative Analysis: If data allows, compare attachment styles and relationship satisfaction across different ethnic or racial subgroups to see if unique patterns emerge in relation to peripartum depression. This could shed light on group-specific vulnerabilities and resilience factors.
- Implications for Practice: Highlight practical outcomes, such as culturally tailored interventions for relationship counseling and mental health support, which could be valuable in clinical settings.
By incorporating these elements, the abstract can showcase a deeper, more nuanced exploration of the topic, which could significantly contribute to both academic and practical fields focused on mental health in diverse communities.
The introduction provides a strong foundation but could benefit from more clarity and conciseness. Here are key suggestions for improvement:
· Emphasize Unique Challenges for WOC: While the text acknowledges the increased risk of perinatal depression among WOC, adding a line on the intersection of racial, socioeconomic, and systemic barriers could deepen this section.
· Clarify Gaps in Literature: Since a major goal is addressing gaps in literature concerning WOC, make this clearer upfront by succinctly outlining what current studies lack and how this study intends to fill those gaps.
The methods section provides a clear quantitative approach, investigating attachment insecurity, mother-infant bonding, and perinatal depression among women of color (WOC). Using regression models, the study aims to clarify whether factors such as relationship dissatisfaction and attachment insecurity predict perinatal depression. Here are some suggestions for improvement:
· The section could benefit from clearer subheadings, especially to differentiate between the various hypotheses and the analytical approach. This could improve readability and flow.
· While each measure is thoroughly detailed, condensing this information might help streamline the focus on relevance, e.g., briefly stating the reliability scores and core use of each measure rather than providing extended descriptions.
· Analytic Strategy Expansion: Consider providing a brief rationale for each step of the hierarchical regression process to clarify the theoretical basis and why the order of variables was chosen in this way.
The results provide a structured exploration of relationship dissatisfaction, attachment insecurity, and their impact on perinatal depression and mother-infant bonding disturbances.
1. Hypothesis 1 confirms that relationship dissatisfaction significantly predicts perinatal depression, with relationship dissatisfaction accounting for 8% of the variance. This finding is statistically robust, showing a moderate negative correlation (β = -.283, p = .002) which aligns with the hypothesis.
2. Hypothesis 2 regarding the prediction of mother-infant bonding disturbances was not supported. Relationship dissatisfaction and attachment insecurity only explained a minimal variance (1-2%) in mother-infant bonding disturbances, with no significant prediction (p > .05). This outcome suggests that these factors may not strongly influence mother-infant bonding within this sample.
3. Hypothesis 3 shows a significant cumulative effect of relationship dissatisfaction and attachment insecurity on perinatal depression. Each variable incrementally explains more variance (up to 34.4% by the final step), with mother-infant bonding disturbances as the strongest predictor (β = .586, p < .001). This hierarchical model effectively maps a progression, indicating that attachment insecurity and bonding disturbances are influential predictors of perinatal depression.
Suggestions for Improvement
1. Clarify Predictive Power in Hypothesis 2: Given that Hypothesis 2 shows weak predictive power, consider exploring alternative variables that may influence mother-infant bonding more strongly within this population (e.g., socio-economic status, support systems).
2. Discuss Practical Implications: Emphasize how findings (especially from Hypotheses 1 and 3) might inform clinical practices, such as relationship counseling or interventions targeting attachment insecurity, to reduce perinatal depression.
3. Address Limitations Explicitly: Mention potential limitations like sample size, the specific demographic (e.g., WOC), and any measurement tools' limitations, which may influence generalizability and predictive strength.
4. Refine the Model Justifications: While the hierarchical model choice is explained, adding a brief rationale for the choice of attachment-related predictors and their assumed impact on perinatal depression would strengthen the narrative.
5. Visual Representation of Results: Consider adding a flowchart or diagram to illustrate the hierarchical regression model and the stepwise variance explained, which can help readers visually track the incremental impact of each predictor on perinatal depression.
Discussion and Conclusion
The discussion effectively highlights the lack of research on WOC (Women of Color), especially regarding relationship satisfaction and attachment bonds during the perinatal period. This clarity establishes the importance and relevance of the study. The section draws well from literature to contextualize findings, particularly around relationship satisfaction and attachment as protective factors against perinatal depression. This foundation strengthens the study’s conclusions. By focusing on how relational dynamics impact mental health, the discussion aligns with real-world implications, advocating for more systemic and relational approaches in therapy for WOC, which is highly relevant and actionable.
The limitations section is thorough, noting issues like potential oversimplification in cultural clustering and the need for qualitative methods. This transparency adds credibility and invites further research. By discussing the importance of cultural and relational factors and noting the absence of inclusive perinatal mental health studies, this section underscores an intersectional approach that enriches the study’s relevance.
The English language used in this article is largely clear and academic, meeting the standards for scholarly writing. Here are some specific observations and suggestions for refinement:
Generally, the grammar and syntax are well-handled, with minor errors. However, some sentences are quite long, which can make them difficult to follow. Breaking these up could improve readability without sacrificing depth. The language is precise and specific, especially when describing research findings and referencing other studies. This precision helps clarify complex ideas for readers. However, a few phrases could be streamlined to avoid redundancy (e.g., "attachment bonds formed with their newborns" could simply be "newborn attachment bonds").
In summary, the English is well-suited for an academic audience but could be polished further by improving readability through shorter sentences, clearer transitions, and minor rephrasing for directness and formality.
